# Non-Functionalized Gold Nanoparticles Inhibit Human Papillomavirus (HPV) Infection

**DOI:** 10.3390/ijms23147552

**Published:** 2022-07-07

**Authors:** Diana Gabriela Valencia-Reséndiz, Atenea Villegas, Daniel Bahena, Kenia Palomino, Jose Manuel Cornejo-Bravo, Liliana Quintanar, Giovanni Palomino-Vizcaino, Luis Marat Alvarez-Salas

**Affiliations:** 1Laboratorio de Terapia Génica, Departamento de Genética y Biología Molecular, Centro de Investigación y de Estudios Avanzados del I.P.N., Av. Instituto Politécnico Nacional 2508, Ciudad de México 07360, Mexico; dgvalr@gmail.com; 2Departamento de Química, Centro de Investigación y de Estudios Avanzados del I.P.N., Av. Instituto Politécnico Nacional 2508, Ciudad de México 07360, Mexico; avillegas@cinvestav.mx (A.V.); lilianaq@cinvestav.mx (L.Q.); 3Laboratorio Avanzado de Nanoscopía Electrónica (LANE), Centro de Investigación y de Estudios Avanzados del I.P.N., Av. Instituto Politécnico Nacional 2508, Ciudad de México 07360, Mexico; dbahenau@cinvestav.mx; 4Laboratorio de Biofarmacia, Faculta de Ciencias Químicas e Ingeniería, Universidad Autónoma de Baja California, Calzada Universidad 14418, Col. Parque Industrial Internacional, Tijuana 22424, Mexico; kenia.palomino@uabc.edu.mx (K.P.); jmcornejo@uabc.edu.mx (J.M.C.-B.); 5Facultad de Ciencias de la Salud, Unidad Valle de las Palmas, Campus Tijuana, Universidad Autónoma de Baja California, Boulevard Universitario 1000, Valle de las Palmas, Tijuana 21500, Mexico

**Keywords:** gold nanoparticles, nanoprophylaxis, viral inhibition, human papillomavirus, HPV, pseudovirus, viral infection inhibition, nanocomplex formation

## Abstract

The spontaneous interaction between human papillomavirus type 16 (HPV16) L1 virus-like particles (VLPs) and non-functionalized gold nanoparticles (nfGNPs) interferes with the nfGNPs’ salt-induced aggregation, inhibiting the red–blue color shift in the presence of NaCl. Electron microscopy and competition studies showed that color-shift inhibition is a consequence of direct nfGNP–VLP interaction and, thus, may produce a negative impact on the virus entry cell process. Here, an in vitro infection system based on the HPV16 pseudovirus (PsV) was used to stimulate the natural infection process in vitro. PsVs carry a pseudogenome with a reporter gene, resulting in a fluorescent signal when PsVs infect a cell, allowing quantification of the viral infection process. Aggregation assays showed that nfGNP-treated PsVs also inhibit color shift in the presence of NaCl. High-resolution microscopy confirmed nfGNP–PsV complex formation. In addition, PsVs can interact with silver nanoparticles, suggesting a generalized interaction of metallic nanoparticles with HPV16 capsids. The treatment of PsVs with nfGNPs produced viral infection inhibition at a higher level than heparin, the canonical inhibitor of HPV infection. Thus, nfGNPs can efficiently interfere with the HPV16 cell entry process and may represent a potential active component in prophylactic formulations to reduce the risk of HPV infection.

## 1. Introduction

Infection by the high-risk human papillomavirus (HPV) is the most significant risk factor in the development of malignant tumors of the anogenital tract, including cervical cancer. Despite the increasing use of prophylactic vaccines against HPV infection, cervical cancer is still a primary health threat for women living in developing countries. Vaccine cost, inadequate healthcare infrastructure and socio-cultural issues are some of the main reasons for this apparent lack of effectivity. Hence, there is still a pressing need for new and efficient strategies for HPV transmission control, detection and treatment [1,2].

Genital dysplasia and cervical cancer are associated with a subset of HPVs referred to as high risk, consisting of HPV types 16, 18, 31, 33, 45, 52 and 58. HPVs are small (50 nm) icosahedral viruses with an 8 kb double-stranded circular DNA genome, encoding eight early and late genes. Early genes regulate cellular transformation, and early and late viral DNA replication. The late genes code for the L1 and L2 proteins, which self-assemble within the host cell nucleus to encapsidate the viral DNA and constitute infective virions. High-risk HPVs infect mucosal genital epithelia, where the key event is the binding of the positively charged viral capsid to heparan sulfate (HS), a negatively charged linear oligosaccharide attached to a protein core on the cell surface [3].

Gold nanoparticles (GNPs) are biocompatible Au clusters from 1 to 100 nm in size. GNPs display distinctive physicochemical properties depending on their size and shape, including surface plasmon resonance (SPR) and fluorescence enhancement effects [4]. Different forms of GNPs have been used as small-volume, ultra-sensitive label-free optical sensors, leading to the development of several biomedical applications, such as real-time imaging and biosensing, as well as the delivery of either genes or anti-tumor drugs for cancer diagnosis and therapy, respectively. An easily noticeable red to blue color shift results from particle–particle plasmonic coupling of GNPs. While in suspension in water, GNPs electrostatically repulse each other, maintaining a stable color. However, the presence of counterions (i.e., Na^++^) neutralizes electrostatic repulsion and the distance between the GNPs decreases, producing near-field coupling that results in bigger particle size and a strong enhancement of the localized electric field within the interparticle spacing, leading to a pronounced color shift. The clearly distinguishable color shift facilitates detection readout to such a degree that it can be performed with the naked eye [4].

GNP conjugation (functionalization) with biomolecules has been extensively reported in several biodetection methods. Non-functionalized GNPs (nfGNPs) in water suspension will eventually aggregate and precipitate as metallic gold powder. However, nfGNP stabilization by protective or capping agents, such as thiols or citrate, prevents uncontrolled aggregation, allowing long-term storage. The capping agent generally forms a monolayer coat (ligand shell) that plays a critical role against aggregation, thus improving solubility and electron-transfer efficiency while allowing controlled shape and assembly. The ability to integrate nfGNPs into biological systems relies on the basic understanding of their interactions with electrolytes, macromolecules and metabolites adsorbed onto their surface by electrostatic, hydrophobic, van der Waals and dispersive forces forming a dense coating known as the biocorona. Such a coating shields the original nfGNP surface properties acting as a “complex” surfactant interface modifying their size and composition, thus defining the physiological response and interaction with living systems including viruses [5].

Our group first reported the spontaneous interaction between HPV16 L1 virus-like particles (VLPs) and nfGNPs, which resulted in the stabilization of purified VLPs and the inhibition of the nfGNPs’ salt-induced aggregation. Ionic-competition and genetic data showed that the nature of an nfGNP–VLP interaction was mainly hydrophobic [6]. Furthermore, nfGNPs covered substantial areas on the VLPs’ surface, suggesting that nfGNPs may be used as a steric inhibitor of the HPV-16 virion interaction with the host cell surface receptor, thus blocking infection. The present work showed, for the first time, the interaction of nfGNPs with functional HPV16 PsVs and characterized the effect of the PsVs in the nfGNPs’ aggregation process and the impact of the nfGNPs in the HPV virion cell entrance process in vitro. Aggregation assays showed that PsVs can inhibit the nfGNPs’ color shift in the presence of NaCl in a dose-dependent manner. High-resolution TEM showed that nfGNPs can directly interact with HPV16 PsVs. The nfGNP–PsV complex prevented the characteristic red to blue color shift in aggregation assays and, in the presence of ethanol, revealed that the interaction is essentially hydrophobic. The PsVs’ aggregate size in solution decreased rapidly in the presence of nfGNPs, as previously determined with VLPs. The nfGNPs were capable of inhibiting virus entry in a pseudovirus model with no toxic effects. This is not unique to nanometric gold, as non-functionalized silver nanoparticles (nfSNPs) can also inhibit the viral cell entry process. We compared the inhibition effect of non-functionalized metallic nanoparticles (nfMNPs) with heparin, the canonical inhibitor of HPV, and showed that nfGNPs can inhibit the infection process more efficiently than heparin.

Non-functionalized metallic nanoparticles (nfMNPs) can be useful to efficiently inhibit the HPV16 infection process and have the potential to be incorporated in prophylactic formulations as a functional component.

## 2. Results

### 2.1. Non-Functionalized Gold Nanoparticle (nfGNP) Characterization

HPV-16 L1/L2 pseudoviruses (PsVs) (a functional model for HPV infection) were incubated with nfGNPs in HPV pseudo-infection assays. The nfGNPs were synthesized by a modified citrate reduction method and characterized by high-angle annular dark-field scanning transmission electron microscopy (HAADF-STEM) as spheroidal shaped particles of about 10–20 nm in diameter (Figure 1A). Dynamic light scattering (DLS) analysis showed a narrow nfGNP size distribution of 18 nm with a standard deviation of 8.164, a polydispersity index of 0.245 and a zeta potential of −22.8 mV with a standard deviation of 6.63, in the synthesis reaction final conditions at pH 5.5 (Figure 1B). These results indicate that the synthesis produced a uniform nfGNP population. For functional characterization, nfGNP response to Na^++^ counterions was tested by incubating 100 µL of nfGNPs (2 nM) with increasing NaCl concentrations (20–100 mM) and measuring the red–blue color shift by absorbance at a 650/520 nm wavelength ratio. Color change was apparent from a 20-mM NaCl increase in the 650/520 ratio, showing that the produced nfGNPs were competent for further experimentation (Figure 1C).

### 2.2. Characterization of PsVs 

HPV16 PsVs expressing the yellow fluorescent protein (YFP) reporter gene were produced in 293TT cells and purified through ultracentrifugation, as previously described [7]. The PsVs were immunoreactive to HPV16 L1 and L2 antibodies in immunoblot assays (Figure 2A). Transmission electron microscopy (TEM) analysis showed homogeneous 55 nm icosahedral capsids (Figure 2B).

### 2.3. nfGNPs Interact with PsVs

As an initial approach to determining the effect of nfGNPs on HPV16 infection, the reported inhibitory effect of HPV16 L1 VLPs on the aggregation of nfGNPs was further tested with HPV16 L1/L2 VLPs and PsVs, which closely resemble the natural virion. Challenging nfGNPs (100 µL) with L1/L2 VLPs or PsVs (200–500 ng) in aggregation assays showed that the inhibition of nfGNP aggregation is not exclusive for L1 VLPs [6] but was also extensive to L1/L2 VLPs (Figure 3A). This was expected because the L1 protein that comprises most of the HPV capsid and L2 is minimally exposed on the virion surface [8]. Moreover, PsVs blocked salt-induced aggregation of nfGNPs, establishing that nfGNP–PsV and nfGNP–VLP interactions are very similar or identical. TEM and HAADF-STEM microphotography showed a direct interaction of nfGNPs with the PsV surface (Figure 3B,C), suggesting that an nfGNP–PsV interaction may have biological relevance to blocking the infection process [9].

As with VLPs, the PsVs tend to form clusters in suspension because of the relative high abundance of hydrophobic contacts on the virion surface [10]. Such contacts are likely destabilized by the hydrophobic interactions occurring with the nfGNPs [6]. The effect of nfGNP concentration on PsV clustering (size about 200 nm) was analyzed using purified HPV16 PsVs incubated with increasing amounts of nfGNPs (50–300 pM) and immediately analyzed by DLS. The nfGNP treatment resulted in the disaggregation of the PsV clusters into particles around 50 nm (Figure 3D), consistent with the previously reported nfGNP-induced disaggregation of HPV16 L1 VLP clusters [6].

### 2.4. nfGNP-PsV Interaction Is Mediated by Hydrophobic Interactions

Toward further insight into the nature of nfGNP–PsV interaction, aggregation assays were performed with rising EtOH concentrations to increase solubility as previously reported for HPV VLPs [6] (Figure 4). The observed recovery of aggregation suggests that, as with VLPs, the nfGNP–PsV interaction is mediated through the participation of hydrophobic interactions.

### 2.5. nfGNPs Inhibit HPV16 PsV Pseudo-Infection

Infection of 293TT cells with HPV16 PsV is the most widely used model for HPV infection. To evaluate the effect of nfGNPs on the HPV16 infection process, HPV16 PsV was treated with increasing amounts of nfGNPs before addition to monolayer 293TT cultures. A qualitative evaluation by fluorescence microscopy of the infected cells showed lower fluorescence levels relative to higher nfGNP concentrations (Figure 5A). Flow cytometry quantification of the pseudo-infected cultures showed that the nfGNP treatment decreased PsV-mediated fluorescence by up to 80–90% relative to the non-treated PsV control. Cell viability assays showed no harmful effects of the nfGNPs on 293TT cells (Figure 5B).

### 2.6. nfGNPs Efficiently Inhibit Pseudo-Infection at Ineffective Concentrations of Heparin

To evaluate whether the inhibition property of nfGNPs on HPV16 infection was restricted to nanometric gold, non-functionalized silver nanoparticles were tested. First, the nfSNP–PsV complex formation was confirmed by microscopy. TEM negative stain showed that nfSNPs interact with PsVs the way as nfGNPs; the nfSNP–PsV complex formation did not disturb the PsV three-dimensional structure visually (data not shown). Second, relative infection was determined to nfSNPs. Flow cytometry quantification of the pseudo-infected cultures showed 80% infection inhibition with nfSNPs in a dose-dependent manner (Figure 6), as previously shown with nfGNPs.

Heparin is a potent HPV16 pseudo-infection inhibitor, as previously reported [7]. To compare with the HPV16 pseudo-infection inhibition potential of nfGNPs, HPV16 PsV was treated with increasing heparin concentrations before addition to monolayer 293TT cultures. Heparin did not show the HPV16 pseudo-infection inhibition effect for the tested concentrations (Figure 6), as previously reported [11].

## 3. Discussion

Pseudo-infection inhibition by nfGNPs was better than by heparin, a molecule structurally related to HS and widely accepted as an efficient inhibitor of the HPV infection [3]. However, the antiviral application of heparin is contraindicated due to its anticoagulant activity and inherent toxicity. On the other hand, nfGNPs have high specificity, even in a complex mixture of proteins, and have a hydrophobic surface that mediates nfGNP–PsV interaction, as recently reported for an influenza virus-sensing platform [11].

Metal nanoparticles (MNPs) have been widely used to inhibit viral infection, and this has been extensively reviewed by Kaminee Maduray and Raveen Parboosing [12]. Functionalized gold nanoparticles with polyethylene glycol (GNP-PEG) of 10 nm in size have been used to inhibit the HIV cell fusion process, supporting the idea of inhibiting the HIV infection process by gp120-CD4 recognition by raising 60% of inhibition at 0.8 mg/mL [13]. Polysulfated gold nanoparticles (GNP-SO_4_Na) from 110 nm were used to efficiently inhibit the binding of vesicular stomatitis virus (VSV) by a three-dimensional network formation that prevents VSV from cell binding; conversely, 19 nm GNP-SO_4_Na only decorate the virion at the concentration tested, probably inhibiting virus-cell binding by a complete steric shielding [14]. nfGNPs from 9.5 nm in size (initially 24 h after infection and every 24 h thereafter, for a total of three times at concentrations of 5 mg/kg and 20 mg/kg) were applied intranasally in 8–10-week-old female BALB/c mice against H2N2 influenza virus, obtaining 100% survival at eight days compared with zero survival for the control [15].

In perspective, nfGNPs could be a component of a novel topical application product as suggested for carrageenan (a polysaccharide such as heparin and a potent inhibitor of HPV infection) and the positively charged polymer AGMA1 [11]. Thus, incorporating nfGNPs into a mixture with carrageenan and AGM1 in a single treatment could completely inhibit HPV entry in different ways, resulting in an effective prophylactic tool. The major concern with nanoparticle application in vivo is its potential toxicity; this concern has been widely reviewed by Sani et al. [16]. nfGNPs have a great potential to be incorporated as a functional component in a lubricant, where the nanoparticles would be in contact with the mucus-coated vaginal tissue—a non-keratinized stratified squamous epithelia. Vaginal epithelium is structured by basal membrane, stratum granulosum, stratum spinosum, intermediate cells, superficial squamous flat cells and stratum corneum cells. Stratum corneum cells, where nfGNP toxicity could occur, slough off by a natural process [17]. Ensign et al. exposed the difficulty of going through the mucous layer coating the vagina epithelium. Mucus traps particles efficiently by adhesive and steric interactions, preventing cell penetration. Particles and pathogens trapped in the upper mucus layer would be expected to rapidly clear [18].

This is the first report of nfGNPs blocking HPV infection. Although a full understanding of the molecular mechanisms governing the nfGNP–HPV interaction remains to be fully understood, the strong inhibitory effect of the nfGNPs on the PsV pseudo-infection process could provide a novel and affordable prophylactic method against HPV infection, thus preventing cervical cancer.

## 4. Materials and Methods

### 4.1. Synthesis of Non-Funtionalized GNPs (nfGNPs)

Non-functionalized GNPs (nfGNPs) were prepared by a modified Turkevich method, as described [19,20]. Briefly, 45 mL of deionized water was added to a reaction flask and refluxed. A mixture of 5 mL of 0.1% tetrachloroauric acid, 2 mL of 1% trisodium citrate and 42.5 μL of 0.1% silver nitrate was added dropwise. Synthesis reactions were carried out for 60 min and allowed to cool to room temperature. Colloidal nfGNPs were stored at 4 °C until used. The nfGNP characterization was made in the synthesis final reaction conditions (pH 5.5), nfGNP shape was determined by HAADF-STEM in a JEM-ARM200F transmission electron microscope (JEOL Ltd., Tokyo, Japan). nfGNP size was determined using dynamic light scattering (DLS) in a Zetasizer Nano-ZS90 (Malvern Panalytical Ltd., Cambridge, UK) with detection at 173°, and nfGNP concentration was determined by UV-visible spectra, using the equation c = A450/Ɛ450, where c is the concentration in mol per liter, A450 is the absorbance at 450 nm and Ɛ450 is the molar extinction coefficient for GNPs of 18 nm (3.87 × 108 M^−1^ cm^−1^) [21]. The nfGNP suspension pH was changed to pH 7.2–7.7 through 1:100 dilution in DPBS buffer before incubation with the PsVs.

### 4.2. Production of HPV16 L1 Virus-Like Particles (VLPs) 

HPV16 L1 virus-like particles (VLPs) were produced as we previously reported [1,22]. Briefly, *Spodoptera frugiperda* Sf21 cells (Life Technologies Corporation, Carlsbad, CA, USA) in Sf900TMII medium (Life Technologies) were infected with L1-producing recombinant baculovirus for 72 h at 26 °C. Infected cells were pelleted by centrifugation and resuspended in 1X Dulbecco’s phosphate-buffered saline (D-PBS) (2.67 mM KCl, 1.47 mM KH_2_PO_4_, 137.93 NaCl mM and 8.06 mM Na_2_HPO_4_-7H_2_O pH 7.2–7.7) complemented with cOmplete^TM^ protease inhibitor cocktail (Sigma-Aldrich, St. Louis, MI, USA) in siliconized microtubes and lysed by sonication in a GEX 130 PB ultrasonic processor (Cole-Parmer Instrument Co., Vernon Hills, IL, USA) at 60% cycle duty 3 times (5 s each). Total lysates were incubated overnight at 37 °C for VLP maturation; digested at 37 °C for 1 h with *Serratia marcescens* nuclease (Sigma-Aldrich) at 0.1% final concentration; and immediately chilled on ice, mixed with a 0.17 volumes of 5 M NaCl and clarified by centrifugation at 2000× *g* for 15 min. The resulting supernatant was purified through CsCl isopycnic centrifugation and extensively dialyzed against 1X D-PBS at 4 °C. VLP production was verified by negative stain in TEM and immunoblotting, and VLP stock was stored in 1X D-PBS at 4 °C until use.

### 4.3. GNP Aggregation Assays

Non-functionalized GNP aggregation was carried out in 96-well microplates by adding nfGNPs (100 μL), VLPs or PsV and NaCl for a 250 μL final volume. Absorbance at 520 and 650 nm was measured immediately using an Epoch™ microplate spectrophotometer (Biotek Instruments, Winooski, VT, USA). The mean and standard deviation of the color-shift 650/520 ratio from each sample were plotted by triplicate.

### 4.4. Dynamic Light Scattering (DLS)

Sample measurements were performed at 25 °C using a Zetasizer Nano-ZS90 (Malvern Instruments, Malvern, UK) with detection at 90°. The hydrodynamic size of the VLPs was recorded as Z-average hydrodynamic diameter (Dh). The plotted data represent the average size of five measurements from the same sample.

### 4.5. Cell Culture

The human embryonic kidney 293TT (ATCC CRL-3467) cells were cultured in Gibco^®^ Dulbecco’s Modified Eagle’s Medium (DMEM) (Thermo Fisher Scientific Inc., Waltham, MA, USA) supplemented with 5% Gibco^®^ fetal bovine serum (FBS) (Thermo Fisher Scientific), 100 IU/mL penicillin, 100 µg/mL streptomycin (PAA Laboratories Inc., Pasching, Austria) and 200 μg/mL Gibco^®^ hygromycin (Thermo Fisher Scientific) at 37 °C and 5% CO_2_.

### 4.6. HPV16 Pseudovirus (PsVs) Production

HPV16 PsVs were produced as previously described [23]. Briefly, 293TT cells were co-transfected with the p16sheLL expression plasmid [11] and the reporter plasmid pSVLYFP using the Lipofectin^®^ transfection reagent (Thermo Fisher Scientific). Cells were harvested 72 h post-transfection, lysed in lysis buffer (DPBS-9.5 mM MgCl_2_ and 0.25% Brij^®^ 58) and incubated at 37 °C for 24 h for maturation. The mature mixture was digested with Benzonase^®^ (Merck KGaA, Darmstadt, Germany) and clarified by centrifugation. Supernatant was placed on an OptiPrep™ gradient (Sigma-Aldrich) for ultracentrifugation at 170,000× *g* for 20 h. The fraction containing PsVs was recovered and filtered through a Sephadex^®^ G25 column (Cytvia, Marlborough, MA, USA) in DPBS-0.5M NaCl. PsV production was verified by immunoblotting and transmission electron microscopy (TEM). Infectivity was a measure of the number of 293TT fluorescent cells generated by 1 µL of viral stock after 72 h post infection. Each fluorescent cell was considered an infectious unit (InU). PsVs were stored at 4 °C until use.

### 4.7. Pseudovirus Infection Inhibition

293TT cells were seeded into 24-well plates (1 × 10^4^/well) and incubated overnight at 37 °C. Infection mixes were prepared with 5000 InU PsVs and increasing concentrations of nfGNPs (Thermo Fisher Scientific) until a final volume of 80 µL of DPBS. Infection mixes were maintained on gentle agitation at room temperature for 20 min and then added to 293TT in 500 μL of DMEM. PsV adsorption was allowed for 20 min before removal of the infection mix followed by addition of 1 mL of pre-warmed DMEM supplemented with 5% FBS. After 72 h, cells were harvested, washed with DPBS and resuspended in 0.5 mL of DPBS for analysis in a FACScalibur flow cytometer (BD Biosciences, San Jose, CA, USA) with a band-pass filter at 530/30 nm (FL1). Excitation was performed with a 488 nm argon laser (1 × 10^4^ cells/read).

### 4.8. Cellular Viability Assay

293TT cells were treated as described in the Section 4.7. Seventy-two hours post treatment, the cells were harvested and an aliquot of the suspension cell was mixed with an equal volume of 0.4% blue trypan (Bio-Rad Laboratories, Hercules, CA, USA). The number of viable cells was evaluated in a TC10 automated cell counter (Bio-Rad Laboratories).

## 5. Conclusions

Nanomaterials used to integrate theranostic tools have become an essential part of health care research and development. In the present work, nfGNPs were tested as inhibitors for the infection of HPV16 (the main etiological agent of cervical cancer) in an in vitro PsV model that recapitulates the HPV16 L1 interaction with HS as an essential step for viral entry to the host cell. As with HPV16 VLPs, nfGNP NaCl-induced aggregation was inhibited by an interaction with HPV16 PsV, suggesting that nfGNPs can inhibit the viral entry process. Moreover, the nfGNPs stably interacted, disaggregating PsV–PsV complexes through direct hydrophobic contacts with the surface of the virions. Pseudo-infection experiments showed that the nfGNPs can significantly inhibit the HPV16 entry process in a dose-dependent manner without affecting cell viability and more efficiently than heparin, a classic inhibitor of HPV infection. A similar effect was observed with silver nanoparticles, suggesting a general mechanism for metallic nanoparticles.

Therefore, nfGNPs could be used as a part of topical formulations to prevent HPV infection on genital mucosa, thus providing a novel prophylactic alternative for the fight against cervical cancer.

## Figures and Tables

**Figure 1 ijms-23-07552-f001:**
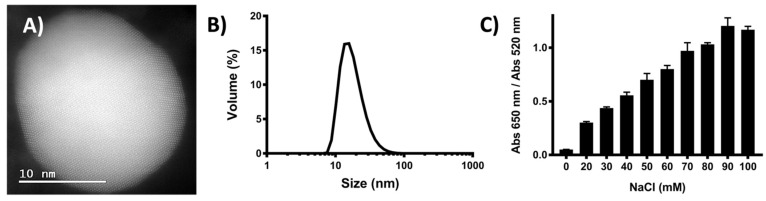
Characterization of non-functionalized gold nanoparticles (nfGNPs). (**A**) Quasispherical nfGNPs were directly visualized by HAADF-STEM (scale bar = 10 nm). (**B**) Hydrodynamic diameter of nfGNPs as determined by DLS. (**C**) Salt-induced aggregation of nfGNPs. Increasing NaCl concentrations were used to determine the minimum concentration required to trigger nfGNP aggregation (color shift from 650 to 520 nm). Bars represent the mean and standard deviation of three independent experiments.

**Figure 2 ijms-23-07552-f002:**
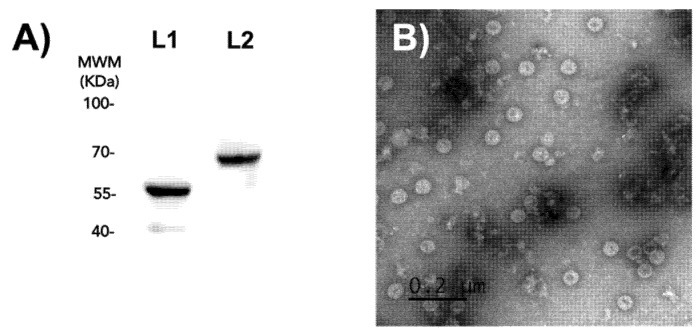
Characterization of PsVs. (**A**) HPV16 PsV immunoblotting. Purified PsVs were electrophoresed by SDS-PAGE and immunoblotted against HPV16 L1 and L2 protein monoclonal antibodies. Prominent 55 and 70 KDa bands corresponding to the L1 and L2 monomer confirmed the purity of PsVs. MWM, molecular weight marker. (**B**) Negative-stain TEM microphotography of PsVs (scale bar = 200 nm).

**Figure 3 ijms-23-07552-f003:**
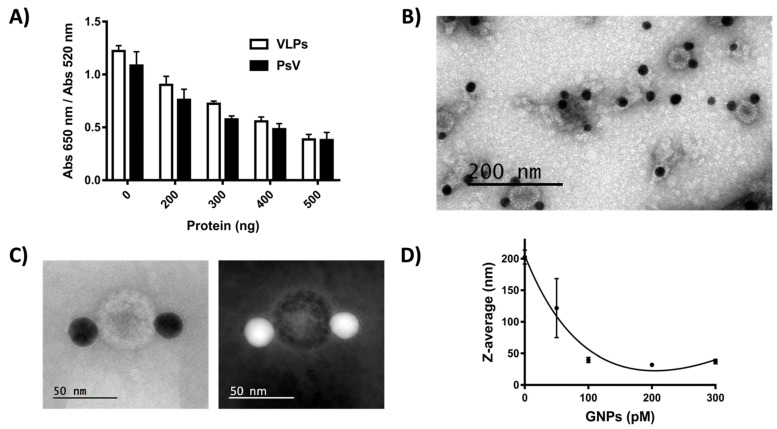
nfGNPs interact with PsV. (**A**) PsV inhibits aggregation of nfGNPs. nfGNPs (200 pM) were incubated with increasing amounts of L1/L2 VLPs (HPV16) and PsV (HPV16) (200, 300, 400 and 500 ng) in the presence of 70 mM NaCl. The graph represents the mean and standard deviation of three independent experiments. (**B**) Negative-stain TEM microphotography of PsV and nfGNPs (scale bar = 200 nm). nfGNPs decorate the surface of PsV. (**C**) Negative-stain HAADF-STEM of PsV and nfGNPs. (**D**) nfGNPs disaggregate PsV clusters. Purified PsV (400 ng) was incubated with increasing concentrations of nfGNPs (50, 100, 200 and 300 pM), and particle size was measure by DLS. The plot represents the average size of five measurements from the same sample.

**Figure 4 ijms-23-07552-f004:**
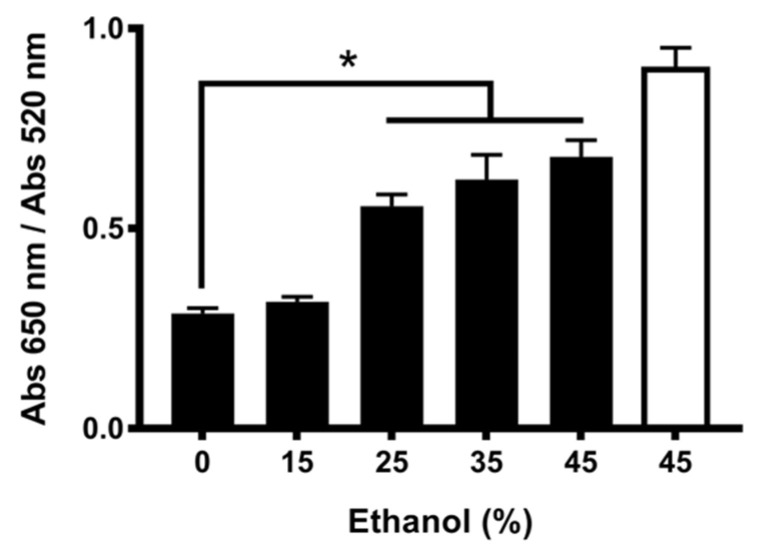
EtOH treatment recovers nfGNP aggregation in the presence of PsV. nfGNPs with increasing concentrations of ethanol were incubated with PsV (500 ng) in 70 mM NaCl. White bar, without PsV; black bars, with PsV. The plot represents the mean and standard deviation of three independent experiments. The asterisk indicates the statistical significance (*p =* 0.0001).

**Figure 5 ijms-23-07552-f005:**
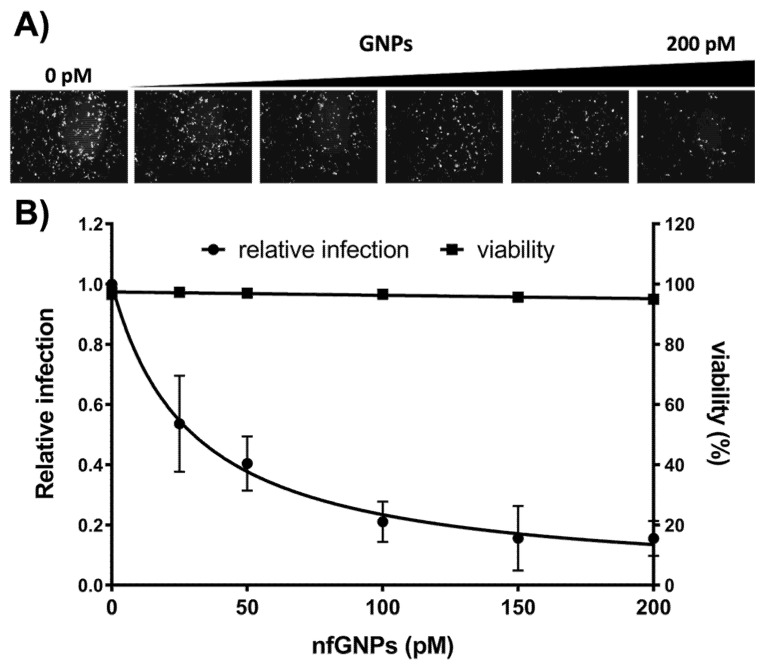
nfGNPs inhibit HPV16 PsV pseudo-infection. (**A**) Fluorescence microscopy of 293TT cells infected with PsV treated with nfGNPs. HPV16 PsV was incubated with increased concentrations of nfGNPs before 293TT cell infection. (**B**) Inhibition of PsV pseudo-infection by nfGNPs. Pseudo-infected 293TT cell fluorescence was quantified by flow cytometry and plotted relative to the non-treated PsV control. The graphs represent the mean and standard deviation of three independent experiments.

**Figure 6 ijms-23-07552-f006:**
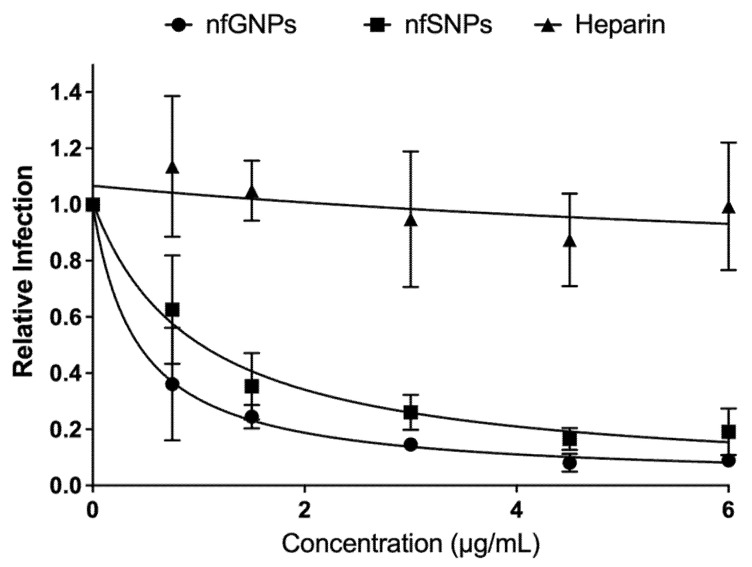
nfGNPs inhibit HPV16 PsV pseudo-infection better than heparin. Inhibition of PsV pseudo-infection by nfGNPs, nfSNPs and heparin. Pseudo-infected 293TT cell fluorescence was quantified by flow cytometry and plotted relative to the non-treated PsV control. The graphs represent the mean and standard deviation of three independent experiments.

## Data Availability

Not applicable.

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
