# Peer review of "Non-Functionalized Gold Nanoparticles Inhibit Human Papillomavirus (HPV) Infection"

_ijms, 2022, doi:10.3390/ijms23147552_

Round 1

Reviewer 1 Report

The authors mentioned that Infection of 293TT cells by HPV16 PsVs is the most widely used model for HPV infection. However, the 293TT are kidney epithelial cells, which is not the target tissue of HPV. Could you justify the model?

Why did you decide to synthesize the nanoparticles and not use the commercial ones?

More information on the characterization of GNPs is needed for a better understanding of the results, as well as for future reproducibility. For instance, the zeta potential, the polydispersity index, and the pH of the solutions used. This is something that should not be underestimated.

The authors do not present the standard deviation or standard error of nfGNPs size distribution of 18 nm (Fig. 1B), and size of icosahedral capsids (Fig. 2B).

The authors claim several times that this is the first report showing that nfGNPs block HPV infection, however, there are other reports. https://pubmed.ncbi.nlm.nih.gov/33688885/ even from the group that is submitting the paper. https://pubmed.ncbi.nlm.nih.gov/28889068/

The word “efficiently” is repeated in line 30-31.

Reviewer 2 Report

The authors have addressed all the raised points and concerns and I recommend publishing their work.

Author Response

thanks for your comment

Round 2

Reviewer 1 Report

Thanks to the authors for addressing the comments. The content of this article is relevant, there is no doubt about that. In addition, now the characterization of the particles was included.

However, there are still a few details to attend in order to improve the manuscript. The standard deviation should be shown to every measured parameter. It is not clear to this reviewer if "standard deviation of 8.164" belongs to the particle size or the zeta potential; and standard deviation for polydispersity index is missing.

Why was the characterization done at pH 5.5? Was this the final pH you worked with? If so, please state this in the methods.

Other than that, the manuscript seems suitable for publicaiton in my opinion.

Author Response

This manuscript is a resubmission of an earlier submission. The following is a list of the peer review reports and author responses from that submission.

Round 1

Reviewer 1 Report

This is a well-designed and written article showing that non-functionalized gold nanoparticles reduce in a high proportion the human papillomavirus (HPV) infection in 293TT cells. Here some suggestions to improve the manuscript.

The authors mentioned that Infection of 293TT cells by HPV16 PsVs is the most widely used model for HPV infection. However, according to the the ATTC, the 293TT are epithelial cells from kidney, which is not the target tissue of HPV. Could the authors provide more details why this is a good model?

Heparin did not show the same or similar HPV16 pseudoinfection inhibition than GNPs at the tested concentrations. However, beyond compare concentrations, would be more interesting to show what concentration of heparin is needed to reach the same effects using nfGNPs. Maybe the authors can include this information based on literature.

It is known that gold and silver nanoparticles induce toxicity in the cells after internalization. GNPs are accumulated due to is not possible to metabolized them and they induce oxidative stress, DNA damage and cytotoxicity (Biochem Biophys Rep. 2021;26:100991.) It is clear that GNPs have a great potential to avoid the HPV infection, but what would happen after the GNPs accumulation in the cells? I encourage authors to include a brief discussion of potential GNPs-induced toxicity.

Why did you decide to synthesize the nanoparticles and not use the commercial ones?

Could you provide more information about the GNPs characterization? For instance, zeta potential, polydispersity index and the pH of the used solutions. In addition, please include the standard deviation of the size distribution.

Minor comments

nfGNPs is defined twice in the introduction. Line 68 and line 82.

Please define the type of cells that is the 293TT cell line.

Reviewer 2 Report

Authors have described a relatively novel approach to managing human papillomavirus (HPV) infection by using non-functionalized gold nanoparticles (nfGNPs) to interfere with the nfGNPs salt-induced aggregation.

I found the work interesting, easy to follow, results are clear and easy to interpret, methods are sound and robust, and the discussion provides reasonable explanations and further work required to confirm the applicability. I rather suggest adding a separate conclusion section or amending discussion section to be Conclusion and Discussion.

I found minor text misspellings that require corrections such as:

Line 61-62 :However, in the presence of counterions (i.e. Na++), neutralizes electrostatic repulsion neutralizes   

Line 96 - 97: The nfGNPs were capable of inhibiting virus entry in a pseu- dovirus model with not (no) toxic effects

Reviewer 3 Report

The authors have performed the synthesis of AuNPs to inhibit the HPV infection. The introduction is well done, easy to follow and with references introducing the reader in the manuscript. Nevertheless, then authors should perform the NPs UV-vis characterization. As well as, they should include in the supporting information the spectrum of all the samples with the increases amount of sodium. Apart from that, I have two comments:

  • When you introduce the gold nanoparticles in DMEM (cell culture medium), they are functionalized immediately by the proteins of the medium. Therefore, when they mix with the cells to inhibit the HPV infection they are not functionalized. It should be important to know how they are functionalized.
  • Second comment is regarding figure 3C. How they know that the gold NPs are attached to the PsVs and it is not a dry effect? Which components have the virus capsules to interact with the NPs?
  • Conclusions are needed.

Round 2

Reviewer 3 Report

Thank you the authors for the response to my questions, nevertheless I still have some comments.

Point 2: When you introduce the gold nanoparticles in DMEM (cell culture medium), they are functionalized immediately by the proteins of the medium. Therefore, when they mix with the cells to inhibit the HPV infection they are not functionalized. It should be important to know how they are functionalized.

Response 1: During the infection experiments, the PsVs were mixed with nfGNPs in Dulbecco's Phosphate Buffered Saline (DPBS) [2.67 mM KCl, 1.47 mM KH2PO4, 137.93 NaCl mM and 8.06 mM Na2HPO4-7H2O pH 7.2-7.7] and incubated by 20 minutes with gentile agitation, to allow for the interaction. After that, the mixtures were added to 293TT cells in 500µL of Dulbecco´s Modified Eagle Medium (DMEM) a synthetic media without proteins (Dulbecco & Freeman, 1959; Eagle, 1959). Then, the nfGNPs-PsVs complexes were incubated with the cells in a monolayer for 20 minutes and the infection mixture was removed from the well followed by the addition of 1 mL of pre-warmed DMEM supplemented with 5% Fetal Bovine Serum (rich in proteins). Thus, the nfGNPs could not be functionalized by the DMEM medium as it does not contain proteins and the nfGNPs were not in contact with the DMEM supplemented with 5% Fetal Bovine Serum.

Comment: Authors have performed the aggregation study until NaCl 100Mm. In the DPBS as they said the concentration of salt is almost 138Mm, besides other salts, the concentration of NaCl is higher than in their experiments. As it can be shown in the figure that they provided in the response the nanoparticles are aggregated at these concentrations, therefore, the role of the nanoparticles without functionalization is uncertain, as they probably are aggregated due to the Schulze-Hardy rule.

In the conclusions (and in the main text, page 6 line 218) is written that the silver nanoparticles could also be used to inhibit the PsVs infection. In the case of silver NPs is well known that they are toxic, therefore, how this toxicity of silver nanoparticles could affect the cells?

Round 3

Reviewer 3 Report

Authors said that there is no aggregation when they mixed the NPs with the DPBS, they observe aggregation with 20 mM of NaCl, therefore, when you have 130 mM you will have higher aggregation. They should show a spectra of the nfGNPs with the DPBS and check time stability. Thats important because when you have your NPs aggregated you lose all the benefits of having gold in nanoscale (that depends on the applications, for their application has no sense to have the NPs aggregated).